# Improved Synthesis of a Novel Biodegradable Tunable Micellar Polymer Based on Partially Hydrogenated Poly(β-malic Acid-co-benzyl Malate)

**DOI:** 10.3390/molecules26237169

**Published:** 2021-11-26

**Authors:** Zhe Yu, Haozhe Ren, Yu Zhang, Youbei Qiao, Chaoli Wang, Tiehong Yang, Hong Wu

**Affiliations:** 1Department of Pharmaceutical Analysis, School of Pharmacy, Air Force Medical University, Xi’an 710032, China; zheyu@fmmu.edu.cn (Z.Y.); zyfmmu163@126.com (Y.Z.); youbei08@163.com (Y.Q.); wangchaoli2012@163.com (C.W.); 2Health Science Center, Xi’an Jiaotong University, Xi’an 710032, China; hren2000@126.com

**Keywords:** crystalline polymer, poly(benzyl malate)(PBM), partial hydrogenation, drug carrier

## Abstract

Poly(benzyl malate) (PBM), together with its derivatives, have been studied as nanocarriers for biomedical applications due to their superior biocompatibility and biodegradability. The acquisition of PBM is primarily from chemical routes, which could offer polymer-controlled molecular weight and a unique controllable morphology. Nowadays, the frequently used synthesis from L-aspartic acid gives an overall yield of 4.5%. In this work, a novel synthesis route with malic acid as the initiator was successfully designed and optimized, increasing the reaction yield up to 31.2%. Furthermore, a crystalline form of PBM (PBM-2) that polymerized from high optical purity benzyl-β-malolactonate (MLABn) was discovered during the optimization process. X-ray diffraction (XRD) patterns revealed that the crystalline PBM-2 had obvious diffraction peaks, demonstrating that its internal atoms were arranged in a more orderly manner and were different from the amorphous PBM-1 prepared from the racemic MLABn. The differential scanning calorimetry (DSC) curves and thermogravimetric curves elucidated the diverse thermal behaviors between PBM-1 and PBM-2. The degradation curves and scanning electron microscopy (SEM) images further demonstrated the biodegradability of PBM, which have different crystal structures. The hardness of PBM-2 implied the potential application in bone regeneration, while it resulted in the reduction of solubility when compared with PBM-1, which made it difficult to be dissolved and hydrogenated. The solution was therefore heated up to 75 °C to achieve benzyl deprotection, and a series of partially hydrogenated PBM was sequent prepared. Their optimal hydrogenation rates were screened to determine the optimal conditions for the formation of micelles suitable for drug-carrier applications. In summary, the synthesis route from malic acid facilitated the production of PBM for a shorter time and with a higher yield. The biodegradability, biosafety, mechanical properties, and adjustable hydrogenation widen the application of PBM with tunable properties as drug carriers.

## 1. Introduction

One of the fastest-growing areas of materials science is the development of biodegradable polymers for biomedical applications [1,2,3]. In particular, biodegradable polymers have been applied in multifarious biological systems as tissue regeneration, medical therapies, and drug delivery [4,5,6,7]. Among them, polyesters represent an attractive family because of their excellent mechanical properties, biocompatibility, and low toxicity [8,9]. In contrast with most common biodegradable polyesters such as polylactides or polycaprolactone, poly(β-malic acid) (PMLA) attracted our attention due to its superior physicochemical properties [10,11,12,13]. The hydrophilicity, plenty of modifiable carboxyl groups, and biodegradability make it a good candidate for drug carrier. PMLA is mainly produced via biofermentation from slime molds such as *Aureobasidium pullulans* and *Physarum polycephalum* [14]. However, obtaining PMLA of a reproducible and controllable molecular weight is difficult to achieve. Direct polycondensation in a one-step reaction from malic acid invariably leads to the heterogeneous poly(malic acid) consisting of α- and β-malate units [15,16]. Currently, the best way to synthesize PMLA is through the benzyl-deprotection of PBM under mild conditions. Hence, an improvement in the yield of PBM is of critical significance.

PBM, unlike PMLA, which can be produced from biofermentation, is most efficiently synthesized via anionic ring-opening polymerization from benzyl-β-malolactonate (MLABn) [17]. The successful preparation of racemic MLABn from l-aspartic acid, which made it possible to synthesize PMLA with high molecular weight, paved the way for the production of this important compound [18,19,20,21,22]. The four-step route in this synthesis pathway provided an overall MLABn yield of 4.5%. Typically, the purification processes needed for MLABn restrain its overall yield. In 1982, the Mitsunobu reaction was first reported to prepare β-lactams from l-malic acid [23]. The latest strategy for the synthesis of MLABn from malic acid was established by Guerin in 1993 [24]. However, further research on the preparation of PMLA from L-malic acid still needs to be improved and refined.

In this study, mlabn was synthesized from l-aspartic acid and d- or l-malic acid. The yield of MLABn obtained from malic acid was 31.2%, which was nearly seven times as much as that obtained from l-aspartic acid and nearly three times as much as before reported from malic acid [18]. A novel crystalline PBM of high optical purity that was polymerized by MLABn was discovered in this study. The crystallization behaviors and thermal properties of the polymers were also determined in detail. Furthermore, the benzyl protection group of PBM was modulated after partial hydrogenation, resulting in an amphiphilic block copolymer that could form nanomicelles of certain sizes for the optimal delivery of hydrophobic drugs via self-assembly in water. These results make PBM a promising drug nanocarrier that could have tailorable properties by controlling its hydrogenation rate.

## 2. Results and Discussion

### 2.1. Preparation of MLABn

In this study, the key product MLABn was synthesized using l-aspartic acid and l/d-malic acid (Figure 1). A controlled molecular weight PMLA made from L-aspartic acid has been previously prepared by our research group [13,24]. The effects of temperature and reaction time on the yield of MLABn prepared using l-malic acid are presented in Appendix A. The yield of MLABn increased first and then decreased with an extension of the reaction time from 6 h to 24 h. As the temperature increased, the intramolecular esterification reaction could not easily proceed because of the instability of lactone. Since the reaction lasted for 12 h at 20 °C, the yield was improved from an earlier report of 11% to 39% in this study [23,25]. Both FTIR and ^1^H NMR were performed to characterize the structure of MLABn. In the FTIR spectra (Figure 2a), the C=O stretching vibrations of the lactone and the benzyl ester group were at 1840 cm^−1^ and 1741 cm^−1^, respectively. As demonstrated in Figure 2a–c, the products MLABn from l-aspartic acid or malic acid were the same. Figure 2f shows that the MLABn synthesized from L-aspartic acid or l-/d-malic acid were correspondingly racemic or optically active. The levorotatory MLABn-2 and dextrorotatory MLABn-3 were obtained with an enantiomeric excess (100%) when prepared from the l-malic acid or d-malic acid enantiomers, respectively. The reaction mechanism of products 1 and 2, as presented in Figure 1, was unimolecular nucleophilic substitution (SN1). Here, the carbocation had a planar structure and negative oxygen ions as nucleophiles attacked both sides of the carbocation at the same probability to obtain racemic monomers. Product 3 has a configuration inversion to obtain a single optically active monomer, which was a Mitsunobu reaction. Hence, the single enantiomer aspartic acid generated racemic MLABn, while the single enantiomer malic acid produced corresponding enantiomer (Appendix A).

### 2.2. Preparation of PBM

The influence of the monomer/initiator ratio and the added sequence of initiator and monomer on the molecular weight of PBM were also investigated. The results showed that the Mw of PBM increased with an increase in the monomer/initiator ratio (Figure 3b). We found that the actual molecular weight of PBM obtained by adding the initiator first was greater than that obtained by adding the monomer first. In the FTIR spectra (Figure 2e), the C=O stretching vibration of lactone at 1840 cm^−^^1^ disappeared after polymerization, indicating a complete reaction. An analysis of the composition of C, H, and O in the two kinds of PBM is shown in Appendix A. The results indicated that there were no obvious differences between these PBM species. The viscosity-average molecular weight ratio of the synthesized PBM-2 is shown in Appendix A.

### 2.3. Crystallization Behaviors of PBM

Different properties between PBM-1 and PBM-2 were obtained as a result of the anionic ring-opening polymerization of MLABn-1 and MLABn-2, respectively. PBM-1 can be found in a transparent semi-solid state that dissolves easily in most organic solvents. In contrast, PBM-2 was a hard white solid that has very poor solubility in many organic solvents. The XRD patterns obtained revealed that the atomic arrangement of PBM-1 was disordered. As illustrated in Figure 4, the obvious diffraction peaks for PBM-2 demonstrated that its internal atoms were arranged in a more ordered and regular manner, which confirmed that it had high crystallinity. The changes in the internal structures of the polymers brought about the variations of the intensities and positions of the FTIR spectra, as shown in Figure 2e. Compared to PBM-1, the peak shape of crystalline PBM-2 was wider, and its intensity at 1750 cm^−^^1^ was weaker. There was a small peak at 779 cm^−^^1^ in PBM-2, which was absent in the amorphous PBM-1. The PBM polymerized from MLABn-3 had the same spectrum as that of PBM-2. It was therefore concluded that PBM-2/3 was polymerized from monomers with mono-optical activity. During the polymerization process of PBM-2, molecular chain interactions, as well as physical crosslinking interactions, became stronger, and the molecular arrangements became more ordered. PBM-1 was polymerized from the racemic MLABn-1, which made the molecules more disorderly arranged during the polymerization process, reducing its crystallinity. The mechanical properties of the crystalline PBM-2 were then characterized by measuring its Young’s modulus. It was found that its Young’s modulus was 573.76 MPa (Appendix A), indicating a fairly high mechanical strength [20,26].

### 2.4. Thermogravimetric Analysis of PBM

The thermal properties and thermal decomposition processes of the synthesized materials were respectively determined via DSC within the range of −20–220 °C at a rate of 10 °C/min and via TG in the range of 35–600 °C at a heating rate of 20 °C/min. As shown in Figure 5, the results of the DSC curve construction indicated that PBM-1 had no crystallization peaks. The glass transition temperature (T_g_) of PBM-1 was 19 °C, as shown in Figure 5a. There was an obvious endothermic peak at 190 °C, which corresponds to the melting temperature (T_m_) of PBM-2. During the cooling process, there were no significant differences between PBM-1 and PBM-2. The reason for this could be that PBM-2 changed from a crystalline to an amorphous state after melting. The thermal decomposition processes of PBM were also determined via TG analysis. According to the TG curves of PBM-1, as shown in Figure 5b, mass loss in PBM-1 started at 233 °C and continued up to 345 °C. As for PBM-2, it began to lose mass at 213 °C until 325 °C. Both TG curves of PBM-1 and PBM-2 showed one-step mass losses of 100%. In order to verify whether PBM-2 changed from a crystalline to an amorphous state after melting, the samples were reheated. As shown in Figure 5c, the DSC curve showed that the melting point of PBM-2 was reduced after reheating. Meanwhile, the glass transition temperature appeared at 16 °C. The crystallization property of PBM-2 vanished after secondary melting and was concluded to have been converted into an amorphous state.

### 2.5. Degradation of PBM

The degradation of PBM was characterized by measuring its degradation rate. The degradation rates of PBM at two temperatures are shown in Figure 6a. After 75 days, the degradation rates of PBM-1 were 42.4% and 71.9% at 37 °C and 50 °C, respectively. As for the crystalline PBM-2, the degradation rate was relatively slow, at 8.2% and 15.1% at 37 °C and 50 °C, respectively, after 75 days. The molecular weight of PBM-1 after degradation at 37 °C for 100 days is shown in Appendix A, and the GPC curves before and after 3 months of degradation are presented in Figure 6b. The surface properties of PBM-1 and PBM-2 were also observed via SEM, and the representative images are shown in Figure 6c. It was observed that PBM-1 had a smooth surface while the PBM-2 showed a more angular and sharp appearance. This could be attributed to the ordered and regular internal atomic arrangement in PBM-2. As the polymer degraded, the surface of PBM-1 became rougher, while the surface of PBM-2 was still angular. At a higher magnification, micropores gradually appeared on the surfaces of the polymers after degradation, demonstrating the effect of their decomposition. The pictures of PBM before and after 3 months of degradation from a macro-perspective are shown in Figure 6e; they were consistent with the above microscopy results [27]. To further illustrate the different degradation behaviors between PBM-1 and PBM-2, their hardness was determined using Moh’s scale [8]. As shown in Figure 6d, the hardness of PBM-2 was between gypsum and calcite, with a hardness value of 143N. By contrast, the hardness of PBM-1 was simply 4.46 N. 

### 2.6. Biocompatibility

PBM-1, which was synthesized from L-aspartic acid, has been shown to be very well biocompatible in a number of previous studies carried out by our group [11,12,13,28]. The crystalline PBM-2 showed disparities with PBM-1. In order to demonstrate the biocompatibility of PBM-2, a skin irritancy test was designed. The results indicated that both PBM-1 and PBM-2 were embedded subcutaneously in rabbits and no erythema or edema was observed after three weeks (Figure 7d,h). HE staining of histologic sections of the subcutaneous tissue demonstrated no acute inflammation or other adverse effects (Figure 7i,j). The cell cycle analysis also proved the biosafety of PBM-2 (Appendix A). These results showed that the new synthesized PBM-2 has good biocompatibility.

### 2.7. Preparation of PMLA

Since crystalline PBM-2 was insoluble in many organic solvents, the synthesis of PMLA from PBM-2 was rarely reported in previous research [17,22,23,29]. In this work, PBM-2 was heated to 75 °C in an oil bath and was hydrogenated to prepare PMLA. PMLA-1 and PMLA-2, which were successfully synthesized from PBM-1 and PBM-2, respectively, were confirmed via ^1^H NMR and FTIR, and the results are shown in Figure 8. Figure 8a shows that the δ_CH_ vibration of the benzyl group at 700 cm^−1^ and 754 cm^−1^ disappeared, and the vibration of OH in the carboxyl group at 3451 cm^−1^ was increased greatly after hydrogenation. Samples for ^1^H NMR were prepared via dissolution in D_2_O (10 mg/mL) at 25 °C. The two proton peaks of the benzyl groups disappeared while the other proton peaks on the main chain remained unchanged (Figure 8b). There were no differences between the structures of PMLA obtained through the hydrogenation of the two kinds of synthesized PBM. The optical rotation values of PMLA-1 and PMLA-2 were −0.847 (c = 0.5 g∙100 cm^−^^3^ in water) and +69.9 (c = 0.5 g∙100 cm^−^^3^ in water), respectively. It was difficult to synthesize PMLA from crystalline PBM because of its insolubility of PBM-2 at room temperature. Ultimately, the differently optically active MLABn-2 and MLABn-3 were mixed together for polymerization to obtain PBM with the same appearance as that of PBM-1 so as to overcome this difficulty.

### 2.8. Partial Hydrogenation of PBM

The partial hydrogenation products PBM_45_H_55_, PBM_23_H_77_, and PBM_17_H_83_ were prepared by controlling the hydrogenation time of PBM. As shown in Figure 9, the diameter of PBM_45_H_55_ was 177.6 nm, and its morphology was less uniform than the other two micelles. The results of the particle size analysis as determined via dynamic light scattering (DLS) revealed that the latter two nanoparticles had hydrodynamic diameters of 111.6 nm and 127.3 nm, respectively. Transmission electron microscopy (TEM) results suggested that the nanoparticles were spherical and were homogeneously dispersed. These results indicated that the partially hydrogenated PBM could self-assemble into spherical micelles in water, with the sizes of the micelles obviously affecting the content of the corresponding hydrophilic blocks created. The CMC value of the blank micelle PBM_23_H_77_ was 0.051 mg/mL. According to our previous research, the π-π stacking between PBM_23_H_77_ and PTX could have contributed to the reduction in CMC values. The diameter of PBM_23_H_77_ @PTX was determined to be 190.9 nm, and its redshift demonstrated this π-π stacking effect. Studies regarding the adjustable hydrophilicity of PBM are of critical importance to create polymeric nanocarrier systems from PMLA that have tunable properties [26,28,30].

## 3. Materials and Methods

### 3.1. Materials

l-aspartic acid was purchased from the Kaiyang biotechnology company (Shanghai, China). l/d-malic acid was obtained from Sigma-Aldrich (St. Louis, MO, USA). Triphenylphosphine and azodicarboxylic acid diisopropyl ester were purchased from Sinopharm Chemical Reagent Co., Ltd. (Shanghai, China). Trifluoroacetic anhydride (TFAA) was purchased from Jinan Wanxingda Chemical company Co., Ltd. (Jinan, China). Sulfuric acid, diethyl ether, benzyl alcohol, tetrahydrofuran, dichloromethane, and dioxane were purchased from Sinopharm Chemical Reagent Co., Ltd. All chemical reagents used were of analytical grade. Palladium/C (Pd/C) catalyst was obtained from Sigma-Aldrich and used without purification. Mouse fibroblast L929 was purchased from the Cell Bank of Type Culture Collection of the Chinese Academy of Sciences (Shanghai, China). PI staining kit and 6-well plates were purchased from TargetMol (Boston, MA, USA).

### 3.2. Measurement

^1^H NMR and ^13^C NMR spectra were obtained on an Avance DMX-500 (Bruker, Rheinstetten, Germany) with tetramethylsilane as the internal standard. Fourier Transform Infrared Spectroscopy (FTIR) spectra were measured by FTIR-8400S (Shimadzu, Japan). UV-vis spectrophotometry was used by UV 6100S. Differential scanning calorimetry (DSC) curve was measured by DSC 214 Polyma (NETZSCH, Selb, Germany) under nitrogen atmosphere. X-ray diffraction (XRD) (2*θ* range 10–80°) was measured on a D8 ADVANCE X-ray diffractometer (Bruker, Rheinstetten, Germany) at room temperature. The thermogravimetric (TG) curve was measured by METTLER TOLEDO System at the heating rate of 20 °C/min under a nitrogen atmosphere. Element analysis was performed by the Vario EL III (Elementar, Hanau, Germany) at room temperature.

### 3.3. Monomer Synthesis

MLABn was synthesized by l-aspartic acid (method 1) and malic acid (method 2).

#### 3.3.1. Method 1

MLABn-1 was synthesized using l-aspartic acid according to our previously published studies [12,18,31]. The purity of MLABn-1 was determined using high performance liquid chromatography (HPLC) with a Diamonsil C_18_ column: 5 μm, 200 mm × 4.5 mm; C_18_ pre-column: 5 μm, 10 mm × 4.6 mm; mobile phase: methanol/acetonitrile/sodium dodecyl sulfate (40:20:40); fluorescence detector detection: excitation wavelength of 254 nm, emission wavelength of 360 nm; flow rate: 0.8 mL/min; injection volume: 5 μL; sensitivity: AUFS = 1.0; and a column temperature of 35 °C. The optical activity of MLABn-1 was also determined via high-performance liquid chromatography (HPLC) using a Chiralcel OD-H column: 5 μm, 46 mm × 250 mm; mobile phase: n-hexane/isopropanol (83:17); fluorescence detector detection: excitation wavelength of 210 nm, emission wavelength of 360 nm; flow rate: 0.8 mL/min; and a column temperature of 25 °C.

The characteristics of MLABn-1 are as follows: yield = 4.5% (with respect to L-aspartic acid), ${\left[\alpha \right]}_{D\;}^{25}$ = +0.024 (c = 1 g·100 cm^−3^ in dioxane), ^1^H-NMR (400 MHz, CDCl_3_): 3.55–3.80 (dd, 2H), 4.85–4.9 (q, 1H), 5.28 (s, 2H), 7.47 (s, 5H); FTIR spectrum (cm^−1^): 11,740 (*v*_C=O_ of COOBn), 1840(*v*_C=O_ of lactone).

#### 3.3.2. Method 2

MLABn-2 and MLABn-3 were synthesized by l-malic acid and d-malic acid, respectively [22,23].

Benzyl-2-hydroxy-3-succinate: trifluoroacetic anhydride (10.7 mL, 75 mmol) was added to (l) or (d)-malic acid (5 g, 37.3 mmol) in an ice bath. After stirring for 3 h at 0 °C (the product was slightly yellow oil), the volatiles were removed by vacuum distillation while the distillation flask was kept at 0 °C (the product became white solid after steaming). The residual solid was dissolved in benzyl alcohol (4.1 mL, 37.9 mmol) with a constant dropping rate and stirred at room temperature for 4 h. The reaction solution was diluted with ethyl acetate (30 mL) and extracted with 10% Na_2_CO_3_. The aqueous phase was combined, acidified to pH 7.0 with 1 mol/L HCl, and extracted with ethyl acetate to remove unreacted benzyl alcohol. The aqueous layers were combined and further acidified to pH 2.0 with 1 mol/L HCl. The aqueous layer was extracted with ethyl acetate; the combined organic phase was washed with saline solution, dried over MgSO_4_, and concentrated to give malic acid benzyl ester (product 3) of 82% yield.

The characteristics of benzyl-2-hydroxy-3-succinate are: (*S*)-benzyl-2-hydroxy-3-succinate ${\left[\alpha \right]}_{D}^{25}$ = −15.4 (c = 0.5 g·100 cm^−3^ in chloroform), (*R*)-benzyl-2-hydroxy-3-succinate ${\left[\alpha \right]}_{D}^{25}$ = +15.5 (c = 0.5 g·100 cm^−3^ in chloroform), ^1^H-NMR (400 MHz, CDCl_3_, δ): 7.31 (s, 5H), 5.21 (s, 2H), 4.54 (dd, 1H), 2.95–2.77 (m, 2H); FTIR spectrum (cm^−1^): 1740 (*v*_C=O_ of COOBn).

Benzyl-β-malolactonate (MLABn-2/MLABn-3): The above monoester (12 g, 0.052 mol) and triphenylphosphine (13.64 g, 0.052 mol) dissolved in 168 mL of dry THF solution were added with DIAD (10.5 g, 0.052 mol) under N_2_ atmosphere in the ice bath. After the mixture was stirred at 0 °C for 30 min, the ice bath was removed, and the reaction mixture was stirred at room temperature for 12 h. The solvent was removed with reduced pressure distillation and obtained a yellow solid, which was then triturated in 100 mL of cold diethyl ether. The white powdery precipitate was removed by filtration. The filtrate was concentrated and purified by chromatography on silica gel with CH_2_Cl_2_: petroleum ether: ethyl acetate (6:6:1) to obtain MLABn (product 2). The synthetic scheme is shown in Figure 1.

The characteristics of MLABn-2 and MLABn-3 are: yield = 31.2% (respect to malic acid), MLABn-2 ${\left[\alpha \right]}_{D}^{25\;}$ = +5.8 (c = 1 g·100 cm^−3^ in dioxane), MLABn-3 ${\left[\alpha \right]}_{D\;}^{25}$ = −8.5 (c = 1 g·100 cm^−3^ in dioxane), ^1^H-NMR (400 MHz, CDCl_3_, δ): 3.55–3.80 (dd, 2H), 4.85–4.9 (q, 1H), 5.28 (s, 2H), 7.47 (s, 5H); FTIR spectrum (cm^−1^): 1740 (*v*_C=O_ of COOBn), 1840 (*v*_C=O_ of lactone). The purity and optical activity of MLABn-2 and MLABn-3 were determined by HPLC as method 1.

### 3.4. Polymerization

MLABn-1 or MLABn-2 and tetraethylammonium benzoate were put into a polymerization tube with a molar ratio of 200:1. The polymerization tube was purged with nitrogen three times, sealed under vacuum, and reacted at 37 °C. The product was dissolved in acetone, added with one drop of concentrated HCl, and poured with a large amount of ethyl alcohol to precipitate the final product PBM. PBM-1 and PBM-2 were polymerized from MLABn-1 and MLABn-2, respectively.

### 3.5. Characterization of PBM

The crystallization behavior of PBM (1 g powder) was characterized via XRD (2θ range, 10–80°) at room temperature. The thermal properties of PBM were measured using DSC spectra at a heating rate of 10 °C/min and TG spectra at a heating rate of 20 °C/min under a nitrogen atmosphere. The morphology of PBM was studied using scanning electron microscopy on Au-coated specimens. It was difficult to detect the molecular weight of PBM-2 via gel permeation chromatography (GPC) because of its poor solubility. Therefore, PBM-2 was dissolved in DMF at 100 °C and cooled to room temperature before further analyses. The molecular weight of PBM-2 was determined using an Ubbelohde viscosity meter, while the molecular weight of PBM-1 was measured via GPC using THF as the mobile phase.

### 3.6. Degradation of PBM

To study the degradation of PBM with different configurations, 1 g of PBM was immersed in a 5 kDa dialysis bag with 300 mL of phosphate-buffered solution (pH 7.4) containing 0.1% *w*/*w* of NaN_3_ at 37 °C or 50 °C, respectively. The samples were dried in a vacuum drying oven after dialyzing for different times. The primitive weight of PBM was denoted as w_1_, and the samples at different times were weighted as w_2_. The mass loss percentage (D%) was calculated as following: D (%) = (w_1_ − w_2_)/w_1_ × 100%.

### 3.7. In Vivo Inflammation Study

The skin irritancy test was carried on rabbits. Two parts of abdominal skin were chosen to test for PBM-1 and PBM-2. After disinfection, an incision was made on each part with approximately 1 cm in length. Then, 0.2 g of PBM-1 and PBM-2 was embedded subcutaneously. After suture and disinfection again, the two parts were observed whether there were erythema or edema after three weeks. Moreover, the nether tissue was fixed with 4% formaldehyde solution, paraffin embedded, and stained with hematoxylin-eosin (HE) reagent for histological examinations.

### 3.8. Cell Cycle Assay

L929 cells (1 × 10^5^ cells per well) were seeded in 6-well plates and incubated overnight. After incubation with free drug medium, the cells were treated with 100 mg PBM-2 for 48 h and fixed with 70% ice-cold ethanol at −20 °C for at least 12 h. After staining, the cells were measured by flow cytometry.

### 3.9. Hydrogenation and Micellization of PBM

Poly(β-malic acid-co-benzyl malate) (PBM_100__−*x*_H*_x_*) was prepared by controlling the hydrogenation time of PBM, where x indicates the percentage of acidic repeating units in the PBM polymer chain. To prepare this, 1 g of PBM dissolved in 20 mL of 1,4-dioxane and 0.4 g of 5% palladium was placed into a 50 mL flask. The system was bubbled with hydrogen and stirred continuously for a certain time at room temperature. The solution was then filtered, the filtrate was condensed via solvent evaporation, and the concentrated solution was mixed into a large amount of ethyl ether to produce PMLA (product 5 in Figure 1). The degree of hydrogenation was determined via ^1^H NMR.

One milligram each of the polymers PBM_45_H_55_, PBM_23_H_77_, and PBM_17_H_83_ (weight average molecular weight, *M*_w_ = 7 kDa) was dissolved in 1 mL DMSO, respectively. The mixtures were slowly added to 1 mL deionized water, and the resulting product was dialyzed using a molecular weight cutoff (MWCO) of 1000 kDa in water for 1 day. The critical micelle concentration (CMC) value was determined using the pyrene fluorescent probe method [32]. A series of PBM_23_H_77_ solutions at concentrations ranging from 6.25 to 250 µg/mL was prepared in which the concentration of pyrene was adjusted to 2.5 × 10^−6^ mol/L. Another series of PBM_23_H_77_ solutions was prepared at concentrations ranging from 6.25–250 µg/mL, which were ultimately dissolved in 2.5 × 10^−6^ mol/L acetone. The fluorescence spectra of the above solutions were recorded using a fluorescence spectrophotometer (F97 Pro, Shanghai, China), with an emission wavelength at 300–450 nm and an excitation wavelength at 335 nm. The CMC value was calculated by plotting the concentration of the polymers versus the ratio of the fluorescence intensities at 373 nm (I_373_) and 383 nm (I_383_).

PBM_23_H_77_@PTX was prepared via dialysis. PBM_23_H_77_ and PTX solutions were dissolved in 4 mL DMSO and then added dropwise into 4 mL deionized water. The final solution was transferred to a dialysis bag (MWCO: 5 kDa) and dialyzed against deionized water at 25 °C for 48 h. The obtained drug-loaded micelles were lyophilized and stored at 4 °C until use.

## 4. Conclusions

In this study, we presented the current state-of-the-art procedures for the synthesis of PBM using optically active malic acid as a starting reagent. The yield obtained was improved nearly seven-fold than previously reported, and the PBM product was synthesized in a more rapid manner. XRD results showed that PBM-2, which was synthesized from L-malic acid, had a crystalline state. Investigation of the thermal properties of PBM-2 further demonstrated its crystallinity and its ability to transform from a crystalline to an amorphous state upon reheating. SEM-imaging analysis revealed the differences in the degradation behaviors between amorphous PBM-1 and crystalline PBM-2. Moreover, we modulated the hydrophilicity of PBM after partial hydrogenation to create an amphiphilic block copolymer and showed that it could form nanomicelles of tunable sizes via self-assembly in water, which can be used for the delivery of hydrophobic drugs. These results provide very useful guiding principles for the rational design of functional biological materials based on PMLA.

## Figures and Tables

**Figure 1 molecules-26-07169-f001:**
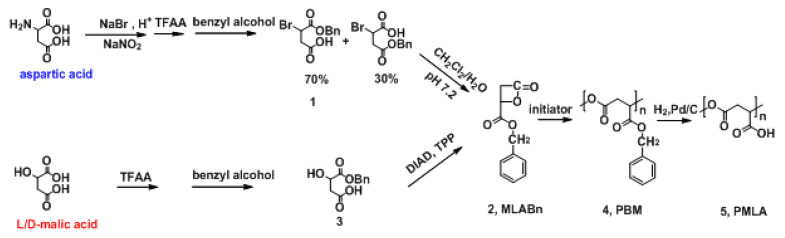
Synthesis routes of PMLA from aspartic acid and L/D-malic acid.

**Figure 2 molecules-26-07169-f002:**
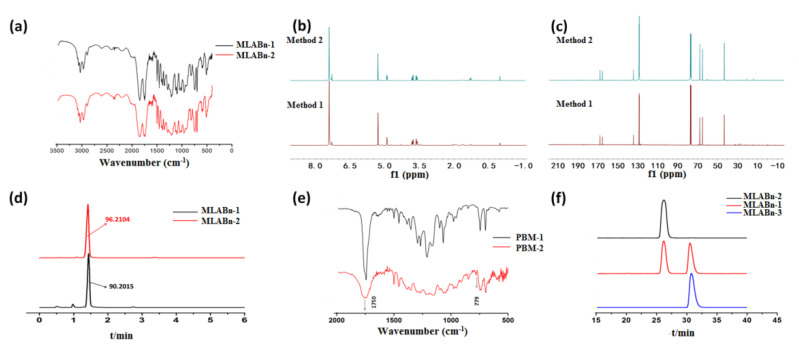
Structural characterization of products synthesized by aspartic acid or l/d-malic acid. (**a**) FTIR spectrum of MLABn; (**b**) ^1^H NMR spectrum of MLABn; (**c**) ^13^C NMR spectrum of MLABn; (**d**) HPLC of MLABn by C_18_ column; (**e**) FTIR spectra of PBM; (**f**) HPLC of MLABn by chiral OD-H column.

**Figure 3 molecules-26-07169-f003:**
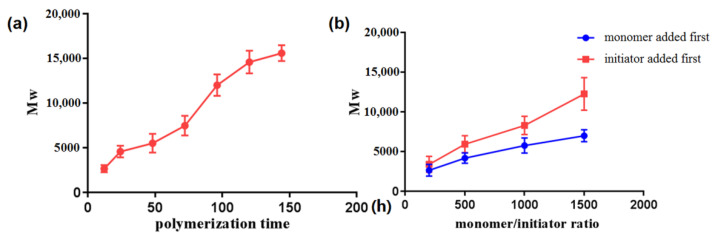
The influence of the polymerization time (**a**), monomer/initiator ratio, and the added sequence of initiator and monomer (**b**) during the polymerization process on the molecular weight.

**Figure 4 molecules-26-07169-f004:**
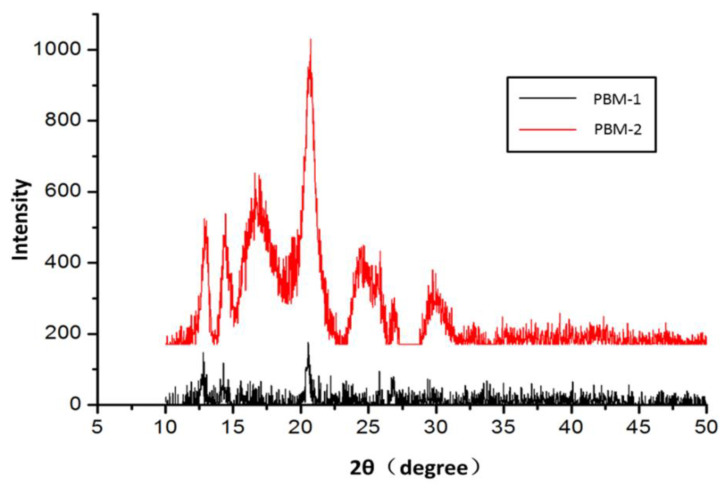
X-ray diffraction spectrum of PBM.

**Figure 5 molecules-26-07169-f005:**
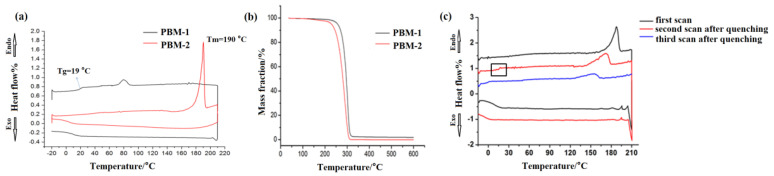
DSC and TG spectra of PBM. (**a**) DSC spectra; (**b**) TG spectra; (**c**) DSC spectra of PBM-2.

**Figure 6 molecules-26-07169-f006:**
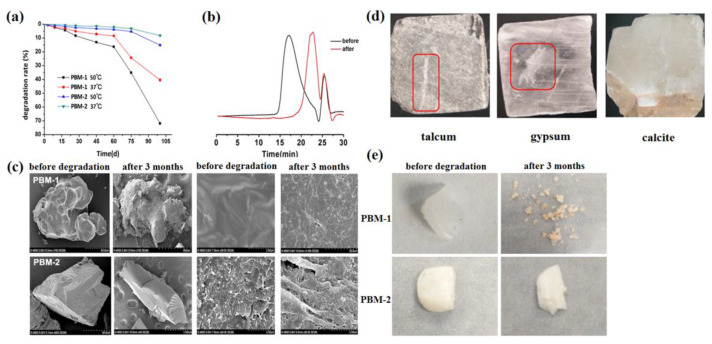
Degradation of PBM. (**a**) The mass loss curves of PBM at 37 °C and 50 °C; (**b**) the GPC curves of PBM-1 before and after 3 months’ degradation at 37 °C; (**c**) the SEM images of PBM before and after 3 months’ degradation; (**d**) the hardness of PBM-2 by scratching on different minerals; (**e**) the real pictures of PBM before and after 3 months’ degradation.

**Figure 7 molecules-26-07169-f007:**
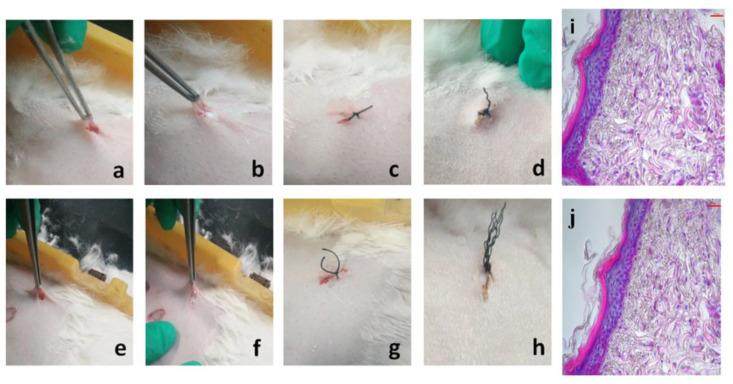
Biocompatibility of PBM. (**a**–**c**) Photograph of PBM-1 embedded subcutaneously; (**e**–**g**) photographs of PBM-2 embedded subcutaneously; (**d**,**h**) degradation subcutaneously after 3 weeks; (**i**,**j**) H–E staining of histological sections of subcutaneous tissue. The scale bar is 20 μm; (**i**) PBM-1; (**j**) PBM-2.

**Figure 8 molecules-26-07169-f008:**
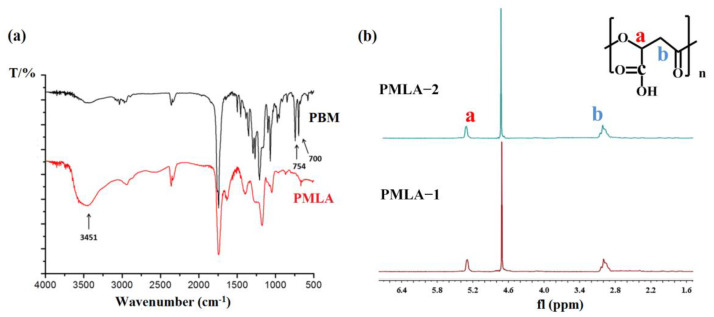
Structural characterization of PMLA. (**a**) FTIR spectrum; (**b**) ^1^H NMR spectrum.

**Figure 9 molecules-26-07169-f009:**
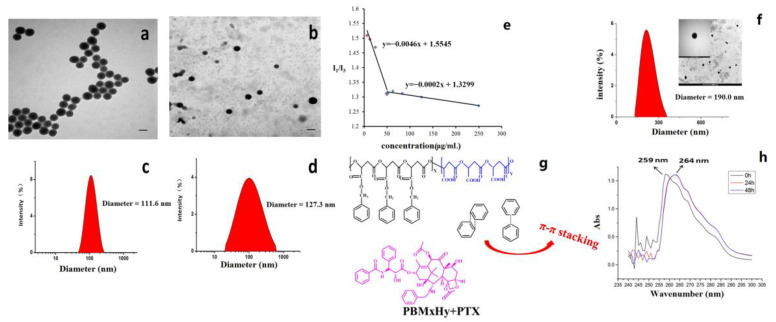
DLS and TEM results of partially hydrogenolyzed PBM micelles. The scale bar is 100 nm. (**a**,**c**) PBM_23_H_77_; (**b**,**d**) PBM_17_H_83_; (**e**) CMC value of PBM_23_H_77_ micelle; (**f**) PBM_23_H_77_@PTX; (**g**) schematic diagram of π-π stacking between PBM and PTX; (**h**) migration of UV absorption curve before and after π-π stacking PBM_23_H_77_@PTX.

## Data Availability

Not applicable.

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
