# Peer review of "Improved Synthesis of a Novel Biodegradable Tunable Micellar Polymer Based on Partially Hydrogenated Poly(β-malic Acid-co-benzyl Malate)"

_molecules, 2021, doi:10.3390/molecules26237169_

Round 1

Reviewer 1 Report

In this work the authors have successfully presented the advanced and controlled synthesis of PBMs from MLABn with an improved yield. The crystalline behaviour, degradation patterns and biocompatibility study of synthesized PBMs suggested their suitability in drug delivery applications. Further, modulation in hydrophilicity of PBMs through hydrogenation to develop amphiphilic block copolymers which form nanomicelles via self-assembly for the delivery of hydrophobic drugs, indicates applicability of PBMs as nanocarriers. However, the manuscript needs some minor modifications which are mentioned below.

  1. In the manuscript some grammatical errors need to be corrected like in Line 40 the word “candidates” should be written as “candidate”.
  2. In Line 24, the sentence “Thus we have tried to………hydrogenated PBM” is not suitable in the context. This should be modified accordingly.
  3. In Introduction part, the description of applicability of various biodegradable polymers in biomedical field could be elaborated for 1-2 lines to provide a solid background to this work and some most recent articles like “10.1016/j.eurpolymj.2020.110155”, “10.3390/pharmaceutics12020095” could be cited.
  4. In some figures (Figure 1a, Figure 4, Figure 5c, Figure 7b, Figure 8b) of the manuscript, the font size of the words and numbers as well as the overall figure size need to be increased.
  5. From Line 84-87, the sentence “HPLC demonstrated that the purity………….higher than MLABn-1” is repeated in the next line, correction is required.
  6. In line 351, w1. should be replaced with w1
  7. In line 352, comma should be replaced from (D, %)
  8. In the figure 6 for biocompatibility studies, authors are advised to add the photograph of the control samples. Here, histological sections of PBM-1 & PBM-2 has been supplied by the author. An elaborative discussion on the relevant subject will be more useful for the readers from various domain.

Author Response

Dear editor and the reviewers,

Thank you for your useful comments and suggestions on our manuscript entitled “Improved synthesis of a novel biodegradable tunable micellar polymer based on partially hydrogenated poly (β-malic acid-co-benzyl malate)” (Submission ID: molecules-1468575). We have modified the manuscript accordingly, and detailed responses to the reviewers’ comments are listed below point by point:

Responses to the reviewers’ comments:
In this work the authors have successfully presented the advanced and controlled synthesis of PBMs from MLABn with an improved yield. The crystalline behaviour, degradation patterns and biocompatibility study of synthesized PBMs suggested their suitability in drug delivery applications. Further, modulation in hydrophilicity of PBMs through hydrogenation to develop amphiphilic block copolymers which form nanomicelles via self-assembly for the delivery of hydrophobic drugs, indicates applicability of PBMs as nanocarriers. However, the manuscript needs some minor modifications which are mentioned below.

  • In the manuscript some grammatical errors need to be corrected like in Line 40 the word “candidates” should be written as “candidate”.

Thanks for the reminder. We have corrected it with red font in the revised manuscript.

  • In Line 24, the sentence “Thus we have tried to………hydrogenated PBM” is not suitable in the context. This should be modified accordingly.

Thanks for the suggestion. The sentence has been revised in order to be more accurate and concise.

  • In Introduction part, the description of applicability of various biodegradable polymers in biomedical field could be elaborated for 1-2 lines to provide a solid background to this work and some most recent articles like “10.1016/j.eurpolymj.2020.110155”, “10.3390/pharmaceutics12020095” could be cited.

Thanks for the suggestion. We have added the information of these recent literatures and made certain expanded elaboration for the applicability of various biodegradable polymers in biomedical field in the revised manuscript.

  • In some figures (Figure 1a, Figure 4, Figure 5c, Figure 7b, Figure 8b) of the manuscript, the font size of the words and numbers as well as the overall figure size need to be increased.

Thanks for the suggestion. The overall figure size and font size have been increased moderately.

  • From Line 84-87, the sentence “HPLC demonstrated that the purity………….higher than MLABn-1” is repeated in the next line, correction is required.

Thanks for the reminder. We have deleted the repeated sentence.

  • In line 351, w1. should be replaced with w1

Thanks for the reminder. We have corrected it.

  • In line 352, comma should be replaced from (D, %)

Thanks for the reminder. We have corrected it.

  • In the figure 6 for biocompatibility studies, authors are advised to add the photograph of the control samples. Here, histological sections of PBM-1 & PBM-2 has been supplied by the author. An elaborative discussion on the relevant subject will be more useful for the readers from various domain.

Thanks for the suggestion. In fact, poly (β-malic acid) and its derivative PBM-1, which is synthesized from L-aspartic acid, have been shown to be very well biocompatible in a number of previous studies carried out by our group. (Y. B. Qiao, et al. J BioMed Nanotechnol., 2019, 15, 28-41; Y. B. Qiao, et al. J Mater Chem B, 2020, 8, 8527; Z. Yu, et al. Polym. Chem, 2020, 11, 7330; Q. Zhou, et al. Theranostics, 2017, 7, 1806.). In the present work, we discovered a crystalline form of PBM (PBM-2) that differed from the previously synthesized PBM-1. Consequently, our initial objective of the experimental design was to utilize PBM-1 as the control to evaluate the variations in biocompatibility of PBM-2, rather than using PBM-1 as a test subject. In any case, we agree with you on the issue that the negative control group could make the results more rigorous and solid, and we will keep your proposal in mind in our subsequent studies. In addition, we have added certain elaborative discussion in the context of the revised manuscript.

All changes have been marked in red in revised manuscript.

Thank you again for your helpful comments and suggestions.

Sincerely,

Hong Wu

Reviewer 2 Report

Journal: Molecules

Title: Improved synthesis of a novel biodegradable tunable micellar polymer based on partially hydrogenated poly (β-malic acidx-co-benzyl malate100-x)

ID: molecules-1468575

This manuscript describes the synthesis of poly(benzyl malate) (PBM) from malic acid through benzyl-β-malolactonate (MLABn) intermediate. In addition, the reaction was compared with the synthesis of poly(benzyl malate) (PBM) from L-aspartic acid. The intermediate compounds and polymers were identified using FTIR and NMR analyses. The polymer was characterized using XRD, DSC and SEM experiments. Moreover, the polymer was investigated for its applicability in the biomedical field. Interestingly, the reported synthetic route provides a improved yield compared to the previous reports. Also, the polymer exhibited compatible characteristics for biomedical applications. The manuscript can be considered for publication after the following minor corrections.

  1. The synthetic schem can be moved from “Material and Methods” section to “Results and Discussion” part for a better understanding of the synthetic discussions.
  2. In Figure 9, when reacting malic acid (37.3 mmol) with excess benzyl alcohol (0.6 mol), only one COOH is protected with the benzyl group. Please explain this.
  3. In title, please change “poly (β-malic acidx-co-benzyl malate100-x)” to “poly (β-malic acid-co-benzyl malate)” for better visibility when searching the article by online.

Author Response

Dear editor and the reviewers,

Thank you for your useful comments and suggestions on our manuscript entitled “Improved synthesis of a novel biodegradable tunable micellar polymer based on partially hydrogenated poly (β-malic acid-co-benzyl malate)” (Submission ID: molecules-1468575). We have modified the manuscript accordingly, and detailed responses to the reviewers’ comments are listed below point by point:

Responses to the reviewers’ comments:
This manuscript describes the synthesis of poly(benzyl malate) (PBM) from malic acid through benzyl-β-malolactonate (MLABn) intermediate. In addition, the reaction was compared with the synthesis of poly(benzyl malate) (PBM) from L-aspartic acid. The intermediate compounds and polymers were identified using FTIR and NMR analyses. The polymer was characterized using XRD, DSC and SEM experiments. Moreover, the polymer was investigated for its applicability in the biomedical field. Interestingly, the reported synthetic route provides a improved yield compared to the previous reports. Also, the polymer exhibited compatible characteristics for biomedical applications. The manuscript can be considered for publication after the following minor corrections.

  • The synthetic schem can be moved from “Material and Methods” section to “Results and Discussion” part for a better understanding of the synthetic discussions.

Thanks for the suggestion. We have shifted Figure 9 to the location of Figure 1 and renumbered the rest of the figures in the revised manuscript.

  • In Figure 9, when reacting malic acid (37.3 mmol) with excess benzyl alcohol (0.6 mol), only one COOH is protected with the benzyl group. Please explain this.

Thanks for the reminder. It was our negligence when preparing the manuscript while the correct amount for the benzyl alcohol should be 37.9 mmol. It has now been corrected in the revised manuscript.

  • In title, please change “poly (β-malic acidx-co-benzyl malate100-x)” to “poly (β-malic acid-co-benzyl malate)” for better visibility when searching the article by online.

Thanks for your suggestion. We have now changed the title in the revised manuscript.

All changes have been marked in red in revised manuscript.

Thank you again for your helpful comments and suggestions.

Sincerely,

Hong Wu
